# Platelet and Thrombophilia-Related Risk Factors of Retinal Vein Occlusion

**DOI:** 10.3390/jcm10143080

**Published:** 2021-07-12

**Authors:** Adrianna Marcinkowska, Slawomir Cisiecki, Marcin Rozalski

**Affiliations:** 1Department of Haemostasis and Haemostatic Disorders, Chair of Biomedical Sciences, Medical University of Lodz, Mazowiecka 6/8, 92-215 Lodz, Poland; maradr@o2.pl; 2Department of Ophthalmology, Karol Jonscher’s Municipal Medical Center, 93-113 Lodz, Poland; cisieckislawomir@gmail.com

**Keywords:** retinal vein occlusion, RVO, platelets, thrombophilia, thrombosis, haemostasis

## Abstract

Retinal vein occlusion (RVO) is a heterogenous disorder in which the formation of a thrombus results in the retinal venous system narrowing and obstructing venous return from the retinal circulation. The pathogenesis of RVO remains uncertain, but it is believed to be multifactorial and to depend on both local and systemic factors, which can be divided into vascular, platelet, and hypercoagulable factors. The vascular factors include dyslipidaemia, high blood pressure, and diabetes mellitus. Regarding the platelet factors, platelet function, mean platelet volume (MPV), platelet distribution width (PDW), and platelet large cell ratio (PLCR) play key roles in the diagnosis of retinal vein occlusion and should be monitored. Nevertheless, the role of a hypercoagulable state in retinal vein occlusion remains unclear and requires further studies. Therefore, the following article will present the risk factors of RVO associated with coagulation disorders, as well as the acquired and genetic risk factors of thrombophilia. According to Virchow’s triad, all factors mentioned above lead to thrombus formation, which causes pathophysiological changes inside venous vessels in the fundus of the eye, which in turn results in the vessel occlusion. Therefore, a diagnosis of retinal vein occlusion should be based on both eye examination and general examination, including laboratory tests.

## 1. Introduction

Retinal vein occlusion (RVO) is a heterogenous disease, in which thrombus is formed the retinal venous system, resulting in the obstruction of venous return from the retinal circulation. The classification of this disorder depends on the site of occlusion: branch retinal vein occlusion (BRVO) and central retinal vein occlusion (CRVO) [1]. The pathogenesis of RVO is still not fully elucidated but is believed to be multifactorial and depends on both local and systemic factors [2,3]. High intra-ocular pressure, glaucoma and short axial distance are regarded as local factors predisposing to RVO [4]. However, hypertension, arteriosclerosis, diabetes, hypercholesterolemia, systemic inflammatory diseases, pregnancy, oral contraceptive use, smoking, hereditary thrombophilia, and increased coagulability are also believed to be risk factors [4,5].

The systemic factors can be further divided into vascular, platelet and hypercoagulable factors, and the vascular factors into dyslipidaemia, high blood pressure and diabetes mellitus. Atherosclerosis and hypertension, which together lead to the modelling a wall of retinal arterial vessels, play key roles in the pathophysiological changes inside venous vessels in the fundus of the eye. The atherosclerotically altered retinal artery exerts pressure on the retinal vein, especially in places when they share a common adventitial sheath. This pressure results in a haemodynamic disturbance in blood flow and, in consequence, to damage to the endothelium. These modifications, according to the assumptions of Virchow’s triad, lead to thrombus formation, and eventually vessel occlusion [2,3,6,7,8,9].

Although arterial and venous thrombosis have been classified as separate phenomena for many years, they share both many similarities in their pathophysiology, as well as their risk factors. Both arterial and venous thrombosis are characterised by a hypercoagulable state, as well as inflammation, the presence of prothrombotic microparticles and excessive platelet activation [10]. The pathogenic mechanism underlying the prothrombotic tendency in retinal vein occlusion is still not fully understood [3,5]. The literature suggests that monitoring selected parameters of blood platelets, such as platelet count, mean platelet volume (MPV), platelet distribution width (PDW), platelet large cell ratio (PLCR) could be useful in the diagnosis of RVO [4,11,12]. It has been observed that excessive platelet aggregation could also be associated with the occurrence of this disease [4]. As regards the plasma component of haemostasis associated with RVO (a potential role of coagulation and anticoagulation systems), there are quite a few scientific reports suggesting a role of both inherited and acquired thrombophilia.

The aim of this review was to present the state of knowledge regarding both platelet- and thrombophilia-related risk factors on the basis of available scientific literature.

## 2. Risk Factors of RVO Associated with Coagulation/Anticoagulation Disorders (Thrombophilia)

A high prevalence of thrombophilia is observed in RVO patients aged less than 45 or 50 years [13,14]. Furthermore, in all patient age groups (≤45 years, 45–60 years, >60 years), the absence of common vascular risk factors, such as high blood pressure, dyslipidaemia or diabetes mellitus, is regarded as the indication of the presence of severe coagulation disorders [13,14]. It was found that the frequency of retinal vein occlusion is higher in patients with previous thromboembolic events elsewhere in the body [13,14]. It has also been noted that a family history of thromboembolism is a particularly important risk factor of thrombophilic disorders in both younger (≤45 years) and older (>45 to ≤60 years) groups of patients [13].

The diagnosis of retinal vein occlusion is based on a combination of eye examination and general examination, as well as laboratory tests. During slit lamp examination, the following changes can be observed in the eye fundus image: vascular tortuosity, retinal haemorrhages, retinal oedema, and cotton wool spots [1]. In addition, it is recommended to examine common cardiovascular risk factors, such as blood pressure, complete blood count, glucose levels, a lipid panel, as well as intraocular pressure in all patients. If the test results are negative, tests for thrombophilia should be performed, particularly in young patients with bilateral eye changes and a positive history of family or previous thrombosis [15].

Thrombophilia can be classified as genetically determined (inherited) or acquired subtypes. Congenital hypercoagulable states are generated as a result of disturbance- mutations leading to a prothrombotic state. They include antithrombin III (AT III) deficiency, protein C deficiency, protein S deficiency, and activated protein C (APC) resistance with the presence of factor V Leiden. Acquired thrombophilia involves elevated antiphospholipid antibodies in antiphospholipid syndrome and hyperhomocysteinemia [15].

### 2.1. Inherited Thrombophilia

Following vascular injury, the coagulation system is activated at the site; this leads to the conversion of fibrinogen to fibrin by a multi-step process, resulting in the formation of a blood clot [16,17]. The haemostasis is regulated by anticoagulant mechanisms, including protein C (PC), protein S (PS), and antithrombin (AT), indicate that a coagulation remains a local process [16,17]. Under physiological conditions, a balance exists between the coagulation and anticoagulation systems. In circulating blood, after conversion into a clot, the activated form of the coagulation factors is inactivated by proteases, one of which is the activated factor V. This is in turn degraded by activated protein C (APC). Any resistance to this inactivation results in an increased risk of thromboembolic complications. The most common origin of APC resistance (95% cases) is a single point mutation in the factor V gene, known as FV Leiden [16,17].

#### 2.1.1. Factor V Leiden Mutation

The genetic risk factor for venous thrombosis is a common mutation in the factor V gene (factor V Leiden) which results in APC resistance [15,18]. Activated protein C resistance is characterized as insufficient anticoagulant response to APC [15]. It has been reported that frequency of the heterozygous variant of this mutation is approximately 2% to 5% in the general European population. Carriers of the homozygous (50–100 fold) and heterozygous forms are believed to be 5–10 fold more susceptible to venous thrombosis than those without any factor V mutation [15,18,19]. Some publications report an association between the presence of CRVO and APC resistance [15,20,21,22,23]. Carriers of the FV Leiden mutation also demonstrate a significantly higher risk of thromboembolic events than those without any mutation in the factor V gene [17].

#### 2.1.2. Protein C Deficiency

Another risk factor of venous and microvascular thrombosis is associated with protein C, which is a vitamin K-dependent serine protease responsible for suppressing the procoagulant system [15,24]. However, while some clinical cases have associated the presence of CRVO with protein C deficiency [15,25,26], others report no significant difference with a healthy follow-up group [15,27].

#### 2.1.3. Protein S Deficiency

Protein S is a vitamin K-dependent glycoprotein playing an important role in the anticoagulation system. Literature data suggest that the frequency of protein S deficiency in patients younger than 45 years with unsolved appearance of venous thrombosis is approximately 5–10% [15,28]. Some reports suggest the presence of protein S deficiency in patients with a diagnosis of CRVO [15,27,29]. However, Pabinger et al. do not report any occurrence of retinal vein occlusion during a 5–10-year clinical prospective study [15,30].

#### 2.1.4. Antithrombin III Deficiency

Antithrombin III (ATIII) is a protein with inhibitory activity towards blood clotting factors such as thrombin and another serine proteases, including factors Xa, IXa, XIIa, active protein C, and kallikrein [15,28,31]. Deficiency of ATIII can lead to severe thrombosis, particularly in peripheral veins and embolism [32]. Patients with known antithrombin III deficiency demonstrate a significant increase in the risk of venous thrombotic events, particularly in young patients (teenagers) [15,31]. Such antithrombin III deficiencies are most frequently inherited as an autosomal dominant disease [15,28]. Associations between antithrombin III deficiency and the appearance of retinal vein occlusion have been reported in some studies but not in others [15,25,32,33,34].

#### 2.1.5. Factor XII Deficiency

Factor XII, also called Hageman factor, is a plasma glycoprotein involved in the initiation of the intrinsic pathway of blood coagulation and fibrinolysis [35]. There are many reports that people with severe factor XII deficiency have a thrombotic tendency [35,36,37,38,39]. It has been found that factor XII deficiency is frequently associated with unexplained primary recurrent miscarriages in women [35,40,41,42,43,44]. Interestingly, several studies have attempted to assess the prevalence of factor XII deficiency in patients with retinal vessel occlusion [33,35,45,46]. Kuhli et al. report that factor XII deficiency is widely distributed among young patients with RVO. Noteworthy, it was observed with similar frequency in older patients with RVO and in healthy individuals [35]. The authors of this work also stressed the fact that factor XII deficiency is a significant risk factor for both central and branch retinal vein occlusion. However, further studies are required to determine the role of factor XII deficiency in retinal vein occlusion [35].

### 2.2. Acquired Thrombophilia

#### 2.2.1. Hyperhomocysteinemia

An increased level of plasma homocysteine is a significant risk factor in venous thrombosis [15,47]. Moreover, hyperhomocysteinemia contributes to the development of atherosclerosis and is associated with classical vascular risk factors associated with arterial and venous thrombosis [14]. High serum homocysteine levels and an increased prevalence of anticardiolipin antibodies have been observed in patients with retinal vein occlusion than in controls [5,48]. A noticeable increase in the level of homocysteine was reported by Lahey et al. among patients with CRVO (10% of the CRVO cases) in comparison to controls [15]. In addition, therapy with folic acid, vitamin B6, or vitamin B12 has a beneficial effect on reducing homocysteine levels and can prevent the venous occlusive disease in patients with high levels, but further investigations are required [15,49].

#### 2.2.2. Antiphospholipid Antibodies

The causes of thrombophilia can be distinguished into acquired conditions, such as antiphospholipid syndrome, and the presence of antiphospholipid antibodies, which are directed against various phospholipid-binding plasma proteins. It is believed that a high level of antiphospholipid antibodies is a significant risk factor for deep venous thrombosis [15,50]. It has been suggested that when retinal vascular occlusion has an unexplained background, antiphospholipid antibodies may play an important role in its pathogenesis [15,51,52,53]. Recently, in a case control study (331 patients with RVO and 281 controls), it was confirmed that the prevalence of antiphospholipid syndrome was higher in RVO patients than healthy subjects [54]. It seems, therefore, that screening tests detecting antiphospholipid antibodies could be useful for informing patients about a high risk of thrombosis, thus allowing early implementation of prophylaxis or even treatment [15,53,55].

### 2.3. Risk Factors of RVO Associated with Thrombophilia—Summary

Elderly people with underlying disorders, such as arterial hypertension, hyperlipidaemia, and/or diabetes mellitus are at particular risk of retinal vein occlusion. These conditions constitute risk factors for atherosclerosis which plays an important role in the pathogenetic mechanisms of RVO [17,56]. A considerably higher incidence of thrombophilia, particularly factor V Leiden mutation, has been observed among subjects without other acquired risk factors. This suggests that the hypercoagulable state rather than changes in the vascular wall is a major factor contributing to thrombosis in this group of patients [17]. Therefore, it is advisable that screening for acquired risk factors, such as hypertension, hyperlipidaemia, and diabetes mellitus should be performed in all patients with RVO [14,17].

Although many studies have demonstrated a relationship between thrombophilic defects and appearance of retinal vein occlusion [35,57,58,59,60,61,62,63,64,65], several others do not report any association [21,66,67,68]. The potential explanation of this discrepancy is a fact that available data have been typically acquired from cases with significant methodologic limitations, such as the retrospective assembly of cohorts, small sample size and an absence of appropriate controls [13,21,57,63,67]. Nevertheless, these studies include tests screening for inherited thrombophilia [5,15] with a special focus on factor V Leiden [17], protein C deficiency [13], protein S deficiency [13], antithrombin III deficiency [13], and factor XII deficiency [35]. Recently, Romiti et al. published a systematic review and meta-analysis assessing a significance of both inherited and acquired thrombophilia in RVO. In total, 95 studies were analysed. The authors concluded that, in comparison with healthy subjects, patients with RVO demonstrated comparable prevalence of inherited and acquired thrombophilias. These results do not suggest that routine screening of thrombophilia should be performed in RVO patients [69].

Risk factors of RVO associated with the presence of inherited or acquired thrombophilia are listed in the Table 1.

## 3. Platelet-Derived Parameters in RVO

The main function of blood platelets is maintaining the integrity of blood vessels through their involvement in haemostasis process [3,70]. Platelets play an important role in many pathophysiological conditions, such as thrombosis, the formation and retraction of thrombus, vessels constriction or inflammatory processes, which promote the development of atherosclerosis in blood vessels [3,71]. In the bloodstream, platelets typically have a discoidal shape. Following activation, they become spherical and grow in size due to the development of pseudopodia [3,72]. Larger thrombocytes contain increased amounts of vasoactive and prothrombotic factors, are characterized by higher expression of adhesion molecules and undergo faster aggregation, which results in more efficient haemostasis and shortened bleeding time [3,73]. In relation to smaller platelets, larger thrombocytes have a higher content of granules, and thus more readily undergo the aggregation process under the influence of activators (agonists) [2]. Moreover, larger platelets contain a greater amount of thromboxane A2 and have an increased expression of glycoprotein Ib and IIb/IIIa receptors [2].

### 3.1. MPV—Mean Platelet Volume

Mean platelet volume (MPV) is a simple blood count parameter that is automatically measured in haematological analysers. MPV reflects the rate of thrombopoiesis but could be also regarded as an indicator of platelet function [2]. Increased MPV values were observed to be associated with platelet hyperactivity [3,74]. In addition, an elevated level of MPV is considered as a risk factor for stroke or myocardial infarction and, interestingly, one study reported an association between MPV and a degree of diabetic retinopathy [3,75,76,77]. The role of MPV was also investigated in several other ocular vascular disorders [78]. A considerable increase in MPV has been observed among patients with symptoms of diabetic retinopathy [77,78]. In addition, a relation has been found between the grade of severity of diabetic retinopathy and MPV values [78]. In another work, Coban et al. reported an association between hypertensive retinopathy and platelet activation [79,80]. In general, literature data suggests that MPV can be regarded as a marker of platelet function, since large platelets have stronger haemostatic reactivity in comparison with platelets of normal size [3,71]. Furthermore, MPV is associated with higher platelet aggregation, increased synthesis of thromboxane, enhanced release of beta-thromboglobulin and excessive platelet adhesion [2,74].

Undoubtedly, platelet activation plays a significant role in thrombosis in the course of symptomatic atherosclerosis [3]. It is believed that elevated values of MPV have prognostic value and are correlated with an incidence of cardiovascular diseases of atherosclerotic aetiology [3]. Elevated MPV levels have been observed in patients having coronary artery disease risk factors such as smoking, obesity, diabetes mellitus, hypertension, and hypercholesterolemia [3,81,82,83,84]. Some data also suggest a correlation between elevated MPV values and the presence of branch RVO [2,3].

Sahin et al. report that patients with symptoms of RVO had significantly higher values of MPV [3,78]. This finding is consistent with those of Onder et al. who demonstrated that MPV was considerably higher in BRVO patients with accompanying hypertension [2]. They also note, however, that further studies on the role MPV parameter as a prognostic biomarker are required [2]. Interestingly, Bawankar et al. showed that MPV was substantially elevated in CRVO patients implying that high values of this parameter correlate with progression of symptoms of this disease [3]. In the recent study by Pinna et al., it was reported that, out of a wide set of parameters based on complete blood count (platelets, leukocytes, neutrophils, lymphocytes, and indices calculated from the above: neutrophil/lymphocyte ratio (NLR), derived NLR, and platelet/lymphocyte ratio (PLR)), MPV was the only parameter significantly higher in the RVO group [85]. By contrast, Ornek et al. observed no association between increased value of MPV and occurrence of RVO [79]. Furthermore, they found that within the group of patients with clinical features of retinal vein occlusion, patients with a diagnosis of branch retinal vein occlusion had a lower value of MPV than patients suffering from central retinal vein occlusion and a control group [79]. The authors suggest that the presence of a structural abnormality in the adjacent arterial vessels could be a determining factor in the development of RVO rather than systemic haemostatic disorders [3,79].

As a conclusion, the majority of literature data suggest that platelet indices such as mean platelet volume reflect platelet function, which is frequently altered in patients with RVO [3]. The prognostic value of RVO was also confirmed in a recently published systematic review and meta-analysis which included 24 studies and 2718 patients with RVO [86]. Nevertheless, further prospective studies are required to fully elucidate the role and significance of MPV as a prognostic biomarker in patients with RVO and to determine the pathophysiology and clinical relevance of elevated MPV value in this group of patients [3]. Finally, it should also be mentioned that recent reports suggest that, beside elevated values of MPV which are observed in RVO patients, some other platelet parameters calculated from complete blood count, such as PDW (platelet distribution width) or PLCR (platelet large cell ratio), could have potential prognostic value for RVO [4] (Table 2).

### 3.2. PDW—Platelet Distribution Width

Attempts to find simple and universal indicators of platelet activation are based on the fact that platelet activation involves morphological changes, such as shape change and formation of pseudopodia. Platelet volume depends on the number and size of their pseudopodia, which probably has an effect on platelet distribution width (PDW) [4,88]. In the recent literature, the consensus is that PDW may be a more unique platelet activation marker than MPV [4,88]. There are reports suggesting that high value of PDW correlates with platelet anisocytosis, which could be associated with formation of platelet pseudopodia [4,88]. Besides, some papers suggest that PDW may have clinical importance: higher values were reported in patients with acute ST-segment elevation [4,89] as well as in patients with diabetes mellitus, especially with associated development microvascular complications [4,87]. Other studies indicate that PDW is independently correlated with the occurrence of undesirable cardiovascular incidents in hospital [4,90] and may represent a self-contained risk factor for cardiac mortality, recurrent myocardial infarction, or the need to conduct further revascularization [4,91]. Yilmaz et al. observed significantly elevated PDW values in retinal vein occlusion patients [4].

### 3.3. PLCR—Platelet Large Cell Ratio

Another parameter routinely recorded during the analysis of complete blood count is the platelet large cell ratio (PLCR). PLCR, similarly to MPV, reflects the proportion of large platelets and platelet volume [4]. A higher value of PLCR can be interpreted as the presence of a high percentage of new platelets characterised by a greater size [4]. It is known that young platelets are larger and more active than mature cells, and contain greater amounts of thromboxane A1, serotonin, and beta-thromboglobulin [4,92]. Literature data indicate that PLCR is inversely proportional to the total number of platelets. However, it is also directly connected with MPV and PDW [4,93]. Elevated PLCR values have been noted in patients with acute coronary syndrome [4,89] and in patients with type 1 diabetes mellitus [4,94]. Significantly increased values of PLCR have been recorded in RVO patients. However, this observation should be confirmed in further studies [4].

### 3.4. Reticulated, Potential New Marker of Platelet Activity

It is believed that the platelet population in the bloodstream is heterogenous with regard to both size and age [95]. The mature forms are accompanied by young platelets, or reticulated platelets, which have been recently produced by the bone marrow as fragments of megakaryocytes [95]. Reticulated platelets can be identified by staining the mRNA [95]. They have a larger number of dense granules and higher mean volume than older platelets [95]. Unlike older platelets, they are also able to synthesise protein due to presence of the residual mRNA [95]. The percentage of reticulated platelets is increased in the circulatory system in conditions of enhanced platelet metabolism, for example during acute coronary syndrome [95]. Hence, it could be speculated that the alteration of this marker can also take place in the pathophysiology of retinal vein occlusion. It seems therefore, that the use of reticulated platelets as a marker of platelet activity is worth considering in the RVO in patients [95].

### 3.5. Activation and Reactivity of Platelets

Although vascular factors, such as hypertension, arteriosclerosis and diabetes are believed to have the predominant impact on the pathophysiology of RVO, the role of abnormal platelet function has also been analysed [78]. Leoncini et al. report enhanced platelet response to thrombin in RVO patients. The authors suggested that the hyperaggregability of platelets, resulting in thrombus formation, plays a key role in initiating and/or contributing to the development of retinal vein occlusion [7,78]. In other studies, increased activation of platelets caused by collagen [9], elevated levels of platelet factors [8], and thrombocyte hyperaggregability [96] has been reported in RVO patients [78]. Kuhli-Hattenbach et al. report considerable platelet hyperreactivity after the induction of very low concentrations of adenosine diphosphate (ADP) in patients with non-arteritic anterior ischemic optic neuropathy (NAION) and retinal vein occlusion without a previous medical history of arterial hypertension, diabetes mellitus, hyperlipidaemia, obesity, and cigarette smoking. This hyperreactivity was considerably greater than in healthy subjects [10]. Notably, ADP is one of the most important physiological activators (agonist) of platelets; it results in the increase and stabilization of thrombus due to thrombocyte activation and strengthens their response to other agonists [10,11,97,98,99,100,101]. The hyperreactivity of platelets manifested even more clearly manifested in the case of patients with a positive family history of thromboembolism [10].

### 3.6. Indices Based on Platelet and Leukocyte Counts

Platelet to lymphocyte ratio (PLR) and neutrophil to lymphocyte ratio (NLR) are increasingly used inflammatory biomarkers that were originally associated with cancer and cardiovascular disease. Due to the low cost and easy method of calculation, they could be regarded as a routine diagnostic element in various disorders [102]. Interestingly, Kurtul et al., in a recent retrospective study of 32 patients with RVO and 32 sex and age matched controls, reported that PLR was significantly increased in the RVO patients, which suggests that PLR could be potential marker of RVO [103]. Similar results were observed for a group (*n* = 81) of BRVO patients where not only PLR, but also NLR values were found to be elevated in the patient group in comparison to control [104]. PLR and NLR markers were also higher in RVO patients as reported in the recent work of Sahin et al. [105].

### 3.7. Platelet Derived Parameters—Summary

Indicators based on simple platelet parameters, such as MPV, PDW, and PLCR provide information on the proportions of large and young forms of platelets [4]. These values are elevated in RVO patients, which is in accordance with the observation that platelet aggregation is enhanced in this disease [4]. It is important to monitor these parameters not only with the eye, but also from a systemic perspective, as RVO constitutes a higher risk of cardiovascular disease [4,106] or impaired heart performance, particularly in patients with accompanied BRVO [4,107]. Patients with RVO also demonstrate a higher risk of cardiovascular disease than a healthy group of a similar age [4,108]. It is known that, similarly to RVO patients, patients suffering from cardiovascular disease have higher values of platelet indices, such as MPV, PDW, and PLCR [4]. This suggests the presence of an association between cardiovascular disease and RVO, and highlights that the progress of these clinical conditions is associated with platelet indicators [4]. It seems, therefore, that platelet indices are positively associated with the risk for the development of cardiovascular disease and cardiac mortality in RVO patients, and they could become tools for their diagnosis and treatment [4]. Importantly, parameters of thrombocytes can be easily measured by performing a complete blood count, which is a very simple and cheap approach [4].

While several reports indicate a relationship between selected hemostasis parameters and the onset of RVO, others do not [20,79,109,110,111,112,113,114]. Trope et al. report some abnormalities in haemostasis and blood viscosity in RVO patients [79,111]: they indicate elevated viscosity of the blood, higher platelet reactivity, and the activation of plasma clotting factors may occur in patients with RVO [79]. It is likely that excessive platelet reactivity may result from higher thrombin generation during coagulation process [79].

On the other hand, Hayreh et al. note that most relevant studies do not suggest any association between haemostatic risk factors and RVO [79,115]. As such, any such cause–effect relationship is unlikely [79]. Similarly, Ingerslev et al. note that the haematological risk factors associated with general venous thrombosis appear only occasionally in retinal vein occlusion, proposing that they play a relatively insignificant role [79,116]. They conclude that there is no specific reason to perform a complete haemostatic examination in patients with RVO [79]. Therefore, it appears that further studies are required to elucidate whether platelets could play a crucial role in the cascade of changes occurring within the pathophysiology of retinal vein occlusion [79].

## Figures and Tables

**Table 1 jcm-10-03080-t001:** Risk factors of RVO associated with coagulation/anticoagulation disorders (thrombopilia).

Risk Factor	Reported Association	No Association Reported
	Inherited thrombophilia	
Factor V Leiden mutation	CRVO: Lahey, 2002 [15], Williamson, 1996 [20], Glueck, 1999 [21], Gven, 1999 [22], Greven, 1997 [23]	
Protein C deficiency	CRVO: Lahey, 2002 [15], Bertram, 1995 [25], Krüger 1990 [26]	RVO: Scat, 1995 [27]
Protein S deficiency	CRVO: Scat, 1995 [27], Prince, 1995 [29]	RVO: Pabinger, 1994 [30]
Antithrombin III deficiency	RVO: Bertram, 1995 [25], Zwierzina, 1985 [32], Hernandez, 2020 [54]	RVO: Guareschi, 1990 [34]
Factor XII deficiency	RVO: Kuhli, 2004 [35]	
	Acquired thrombophilia	
Hyperhomocysteinemia	CRVO: Lahey, 2002 [15]	
Antiphospholipid antibodies	RVO: Fong, 1993 [51], Giorgi, 1998 [52], Dunn, 1996 [53]	

Number in parentheses denotes the number of the reference in the bibliography.

**Table 2 jcm-10-03080-t002:** Platelet-derived parameters in selected ophthalmology diseases.

	MPV	PDW	PLCR
BRVO	I [2]/D [79]	–	–
CRVO	I [3]/D [79]	–	–
RVO	I [78,85]/D [79]	I [4]	I [4]
Diabetic retinopathy	I [77]	I [87]	–
Hypertensive retinopathy	I [80]	–	–

I, increased values; D, decreased values; – no data. Number in parentheses denotes the number of the reference in the bibliography.

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
