# Peer review of "Platelet and Thrombophilia-Related Risk Factors of Retinal Vein Occlusion"

_jcm, 2021, doi:10.3390/jcm10143080_

Round 1
Reviewer 1 Report
The authors to present the state of knowledge regarding both platelet and thrombophilia, related risk factors based on available scientific literature as risk factor of Retinal Vein Occlusion.
Major concerns
#1 In the paper makes a bibliographic review of the different risk factors in VOR without providing data on the articles analyzed, patients who have participated in said studies, or clear conclusions about them.
#2 Little current analyzed bibliography
#3 Table 1: The authors of the articles are named, unclear about the types of studies and results.
#4 Table 2: Little information and not clear.
#5 There is talk of future studies, but is there nothing published on the subject recently?
Reviewer 2 Report
In this paper, Marcinkowska et al. reviews platelet and thrombophilia-related risk factors of retinal vein occlusion.
1. Title: Please remove the abbreviation of RVO in the title.
2. Throughout the manuscript which I have read in PDF, there are a lot of embedded links which refer the reader to a webpage "context.reverso.net" which translates phrases in English-Polish. Please remove such links.
3. There are an important number of language issues throughout the manuscript, which also includes unnecessary convoluted sentences. The manuscript would benefit from a professional language revision.
4. Avoid abbreviation to RP, which is traditionally used in ophthalmology as an abbreviation of retinitis pigmentosa.
Round 2
Reviewer 1 Report
The comment provided were not to solve